# Changes in Body Mass Index, Energy Intake, and Fluid Intake over 60 Months Premortem as Prognostic Factors in Frail Elderly: A Post-Death Longitudinal Study

**DOI:** 10.3390/ijerph17061823

**Published:** 2020-03-11

**Authors:** Yoshiaki Kawakami, Jun Hamano

**Affiliations:** 1Department of Nursing, Tokyo Ariake University of Medical and Health Sciences, Tokyo 135-0063, Japan; 2Division of Clinical Medicine, Faculty of Medicine, University of Tsukuba, Tsukuba 305-8575, Japan; junhamano@md.tsukuba.ac.jp

**Keywords:** body mass index, prognosis, frail elderly, energy intake, death, end-of-life care

## Abstract

Survival prediction is considered difficult in elderly individuals with cognitive frailty or dementia that leads to death. The aim of this study was to verify temporal changes in body mass index (BMI), energy intake, and fluid intake measured continuously in frail elderly people as prognostic factors for death. We assessed 106 frail elderly individuals who received >2-year care at an elderly care facility and died at the facility. We analyzed temporal changes in BMI, energy intake, and fluid intake from a maximum of 60 months premortem to death and determined the relationship between these parameters. BMI was significantly below the reference values from 9 months prior to death, but energy intake remained constant from 2 months prior to death to death (*p* < 0.001). However, the mean fluid intake decreased suddenly immediately prior to death. We compared the changes in each parameter during the year prior to death and during the preceding year and found significant differences in all parameters (*p* < 0.001). From 60 months prior to death to death, BMI, energy intake, and fluid intake decreased in the same order over time. Therefore, these parameters can be used as prognostic factors for death in frail elderly people.

## 1. Introduction

Prevalence of cancer-related death is the highest in individuals aged <80 years; however, elderly individuals who escape death from cancer and organ failure die due to cognitive frailty (senility) and dementia [1]. It is difficult to predict when elderly individuals with cognitive frailty are approaching death [2].

Several predictive tools have been developed to predict survival in individuals with dementia to predict death [3,4,5,6]. The parameters used to predict survival include the presence or absence of disease such as heart failure and respiratory disease, level of dependence and cognitive dysfunction in daily life, malnourishment, presence or absence of bedsores, levels of hemoglobin and serum albumin, and presence or absence of constipation. However, these parameters require a medical specialist and blood testing. Regarding frail elderly individuals residing at home with limited medical care and those receiving medical care at elderly care facilities, it is difficult to collect data on these parameters in a routine and consistent manner.

Dependency for activity of daily living (ADL) can be measured by anyone, and it is an effective parameter to estimate the prognosis of patients with cancer who exhibit a sudden deterioration in ADL immediately prior to death [7,8]. Lynn noted that the trajectory leading to death in patients with cancer or organ failure can be demonstrated and easily understood based on changes in physical functioning [9]. However, in patients who die from cognitive frailty and dementia, ADL and physical functioning deteriorate significantly before death approaches [10], and such changes occur gradually. Therefore, it can be difficult to predict life expectancy based on dependency for ADL alone. Brown reported that despite many reports predicting 6-month mortality in elderly individuals with advanced dementia, no reliable indicators have been established to date [11]. Therefore, in this population, parameters that can be easily and routinely obtained and can predict survival are needed.

Body mass index (BMI: weight (kg)/height (m^2^)) can be easily measured and it has been found that the risk of death is associated with a lower BMI [12,13]. Veronese et al. found that BMI is the most appropriate nutritional predictive factor for long-term mortality in residents of nursing homes and that a lower BMI indicates a higher risk of death [14]. In addition to BMI, which is a long-term predictive factor, additional prognostic factors would be useful for a more accurate prediction of life expectancy.

The aim of the present study was to clarify the changes in BMI, energy intake, and fluid intake among elderly individuals residing in Japanese nursing homes who died without receiving artificial nutrition and hydration (ANH). By examining the point in time when each parameter significantly changed as death approached and the amount of change from before to after this point in time, we evaluated whether these parameters could serve as effective prognostic factors for predicting time until death.

## 2. Subjects and Methods

The present study included all frail elderly people who died during the 6-year period from May 1, 2011, to April 30, 2017, at a Nursing Home located in a typical Japanese Province. This Nursing Home provides specialized care for elderly individuals aged ≥65 years with moderate-to-severe or greater impairment of ADL and cognitive frailty. They were officially recognized to be in a state of requiring long-term care according to the Public Nursing Care Insurance Law. Residents of a facility covered by the Public Nursing Care Insurance, such as this Nursing Home, are required to be officially recognized as being at care-need level 3 or above as per the Long-Term Care Insurance system. A typical example of an elderly individual at care-need level 3 is someone who cannot stand or walk with their own strength and requires full assistance for performing daily activities, such as using the toilet, bathing, and dressing, with some problematic behaviors and poor understanding [15]. All the subjects in the present study were considered to be frail elderly individuals [16].

The subjects included elderly individuals who continuously resided in the Nursing Home for at least 24 months and who died during their stay without receiving ANH. The exclusion criterion was frail elderly individuals who died after being hospitalized for acute diseases requiring continuous medical care that the nursing home could not provide.

Among the 187 individuals who died while residing at the Nursing Home during the survey period, 106 met the inclusion criteria and were included in the present study. The study was planned in accordance with the Declaration of Helsinki, and the study protocol was approved by the Ethical Review Board of Tokyo Ariake University of Medical and Health Sciences (Project identification code: 256; date of approval: July 26, 2018). Data were collected in accordance with the privacy policy of the Nursing Home and consent was obtained from all subjects and/or their legal representatives in advance. Use of data was allowed on the condition that the subjects remained unidentifiable.

### 2.1. Measurements

When the enrolled subjects died, we retrospectively collected the following data from electronic patient records for a maximum of 60 months prior to death.

### 2.2. Physical Measurements

Nurses and trained caregivers measured the height to 0.1 cm and weight to 0.1 kg using a standard protocol [17] at admission and every month thereafter. For those with difficulty in standing up straight, height was measured to 0.1 cm using a measuring tape and weight was measured using a wheelchair scale.

BMI was calculated by dividing body weight (kg) by height (m^2^).

### 2.3. Energy Intake and Water Intake

Meals were prepared for all residents based on the energy intake (kcal) and nutrients recommended by a registered dietician. Upon completion of each meal, caregivers observed and recorded the proportion of meal consumed on a scale of 1–10.

Apart from the fluid content included in meals, caregivers recorded the total amount of fluid intake per day (mL) based on the volume of a cup measured in advance with which the elderly individual consumed water.

Energy intake (kcal) was calculated daily by multiplying the proportion of the meal consumed by the nutritional value of each meal. The amount of fluid intake consumed before or after meals was obtained daily (mL).

### 2.4. Analysis

A total of 106 subjects resided in the nursing home for at least 24 months, among whom 37 resided for ≥60 months prior to death. We calculated the mean value of each parameter immediately prior to death dating back to 60 months prior to death and 1 month prior to death. We calculated the mean BMI measured each month, and the mean energy intake and fluid intake per day of each month when BMI was measured. The changes and relationship between the three parameters were determined.

As a check of the accuracy of such measurements, we compared the relationship between BMI and energy intake. The World Health Organization classifies adults aged ≥20 years with a nutritional status of BMI <18.5 kg/m^2^ as underweight. We calculated the basal metabolic rate based on the mean height and weight of the subjects for 24 months prior to death using the Ganpule equation. This equation estimates the basal metabolic rate of Japanese individuals more accurately than the revised Harris–Benedict equation [18,19]. The mean basal metabolic rate of the 106 study subjects was 830 kcal. Using a BMI of 18.5 kg/m^2^ and energy intake of 830 kcal as reference values, we tested the population mean (one-sample *t*-test) for the study period when the values were significantly below reference values; this was done to determine the changes in BMI and energy intake as well as the relationship between them.

### 2.5. Statistical Analysis

We subsequently calculated the change in each parameter during the year prior to death and during the year starting from 24 months prior to death based on their mean values using Welch’s *t*-test. The predictive ability and cutoff values of all parameters were calculated by ROC analysis to determine the change that occurs as death approaches. Youden’s index was used to determine the cutoff values. Using a logistic regression analysis, we examined the strength of the relationship between the amount of change in each parameter and death.

A *p*-value of <0.05 was considered significant. All statistical analyses were performed using JMP Pro 14.0.0 (for Windows 10) (SAS Institute Inc., Cary, NC, USA).

## 3. Results

### 3.1. Characteristics of the Study Population

All 106 study subjects were Japanese, and 84% of them were female (Table 1). Their mean age at death was 90.1 ± 7.1 years. At the time of death, 72% subjects were aged 85–99 years.

### 3.2. Changes in Each Parameter from 60 Months Prior to Death and the Relationship among the Parameters

The mean BMI was measured each month and the mean energy intake/day and fluid intake/day for the months when BMI was measured were determined for each subject from immediately prior to death to 60 months prior to death. The trends for the three parameters and the relationship among them were determined (Figure 1).

The mean BMI 60 months prior to death was 20.3 kg/m^2^, which decreased over time. A BMI of <18.5 kg/m^2^, considered underweight, was reached on an average 17 months prior to death, and it continued to decline in an irreversible manner, reaching a mean BMI (±standard deviation) immediately before death of 16.2 ± 3.5 kg/m^2^. The mean energy intake/day gradually decreased and reduced below the basal metabolic rate of 830 kcal 2 months prior to death. The mean fluid intake suddenly decreased immediately prior to death. The mean BMI decreased before mean energy intake decreased, which occurred before the mean fluid intake decreased.

We conducted a population mean test (one-sample *t*-test) to determine the month prior to death when the mean BMI was significantly <18.5 kg/m^2^ and the mean energy intake decreased to <830 kcal (Table 2). BMI was significantly <18.5 kg/m^2^ 9 months prior to death, and the mean energy intake was significantly <830 kcal 4 months prior to death.

### 3.3. Relationship between the Amount of Change in Each Parameter and Death

The mean change in BMI from 12 months premortem to immediately prior to death (Δ_1_) and from 24 months to 12 months premortem (Δ_2_) were 1.9 ± 2.0 and 0.9 ± 1.8 kg/m^2^, respectively. The change in energy intake for the corresponding two periods was 648 ± 382 and 49 ± 204 kcal, and the change in fluid intake was 199 ± 243 and −12 ± 191 mL, respectively (Table 3). We found a significant difference in the amount of change for all parameters from 12 months prior to death to immediately prior to death and from 24 months to 12 months prior to death.

The ROC curves for each parameter are presented in Figure 2. AUC was 0.645 for BMI, 0.912 for energy intake, and 0.775 for fluid intake with a significant difference observed for each parameter. The cutoff values measured using Youden’s index (Table 4) were 1.7 for BMI, 300 for energy intake, and 126 for fluid intake (Table 4). Using the logistic regression analysis, the odds ratios per standard deviation were 1.80 for BMI, 17.32 for energy intake, and 3.47 for fluid intake.

## 4. Discussion

In the present study, including 106 Japanese elderly individuals who entered and died in the nursing home without receiving ANH, we retrospectively elucidated the mean progression of BMI, mean energy intake, and mean fluid intake from 60 months (5 years) prior to death.

Because a low BMI increases the risk of death [20,21,22,23], it is an effective factor in predicting the risk of death [24,25,26]. A BMI of 20.0 kg/m^2^ is considered a valid threshold for determining a high risk for death in elderly individuals [11,14,27]; however, the relationship between temporal changes and prognosis has not been studied. In the present study, BMI decreased 60 months premortem and at approximately 17 months premortem; the mean BMI of the study subjects was <18.5 kg/m^2^, which is classified as underweight. During this period, BMI significantly and irreversibly decreased during 9 months premortem.

BMI can be easily measured and longitudinally monitored; thus, it can serve as an effective prognostic factor. Physical constitution differs depending on ethnicity, and a valid threshold for each ethnic group must be determined. It appears that in Japanese people, when BMI is <18.5 kg/m^2^ and decreases by 1.7–1.9 kg/m^2^ over 1 year (approximately 10%), it can be used as a valid marker for determining a high risk of death.

The relationship of the changes in BMI with changes over time in energy intake and fluid intake levels observed in the present study differed significantly during the year prior to death and during the preceding year. The predictive ability and odds ratio increased for the measured parameters in the order of energy intake, fluid intake, and BMI.

Furthermore, while the subjects exhibited a continuously significant and irreversible decrease in the mean BMI during 9 months premortem, the mean energy intake significantly decreased immediately prior to death. Generally, a decrease in BMI is attributed to decreased energy intake or increased relative energy expenditure due to chronic cough and increased activity. However, among the subjects in this study, a decrease in BMI preceded a decrease in food intake, suggesting that the observed BMI decrease is attributable to factors other than decreased energy intake.

We found that BMI decreased despite consuming meals prepared above the basal metabolic rate, indicating that maintaining energy intake does not necessarily lead to continued survival. Frail elderly individuals are not necessarily unable to eat when approaching death, supporting studies indicating that ANH does not necessarily prolong life [28,29].

The calculation of BMI only requires height and weight; therefore, an increase in the volume of body fluid affects BMI. Monitoring the proportion of meal intake serves as alternative means of measuring energy intake, and both intakes are effective parameters for predicting death.

We believe that monitoring shifts in fluid intake over a time period of at least 1 year is an effective parameter to predict death. The study subjects could consume a relatively constant amount of fluids until immediately prior to death by receiving assistance with fluid intake with meals. If the amount of fluid intake suddenly decreases despite assistance, death is essentially unavoidable.

Measurements such as body weight, meal intake, and fluid intake can be collected during at-home care without specific measuring tools. The assessment of these easily obtainable parameters might help to determine whether elderly individuals are in in the terminal phase of frailty, and it might aid in improving end-of-life care. Furthermore, it can provide an opportunity for family members to prepare emotionally for the impending death of an elderly individual.

The present study has certain limitations. First, the sample size was limited to Japanese people from a single nursing home. Second, the vast majority of subjects were female. Lastly, the subject sample was limited to 106 individuals and may not be representative of the whole population. Although we defined frail based on the Long-Term Care Insurance system in Japan, it is unclear whether all residential individuals met the clinical criteria of frail. Therefore, caution is needed while interpreting the results of our study.

In terms of strengths, temporal changes in each parameter were measured for up to 60 months in the present study, resulting in a more reliable prediction.

## 5. Conclusions

BMI, energy intake, and fluid intake may be prognostic factors for death in frail elderly individuals. By longitudinally monitoring these parameters and analyzing the amount of change over time, a more accurate prediction may be possible. Although changes in these parameters are gradual in this population, monitoring these parameters over mid-to-long-term can help record these changes accurately and lead to better end-of-life care.

## Figures and Tables

**Figure 1 ijerph-17-01823-f001:**
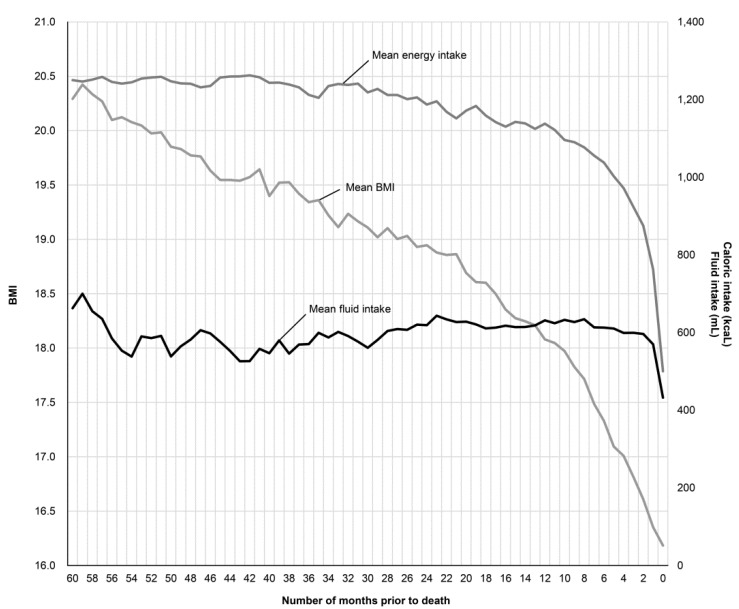
Change in the mean body mass index (BMI), energy intake, and fluid intake 60 months prior to death.

**Figure 2 ijerph-17-01823-f002:**
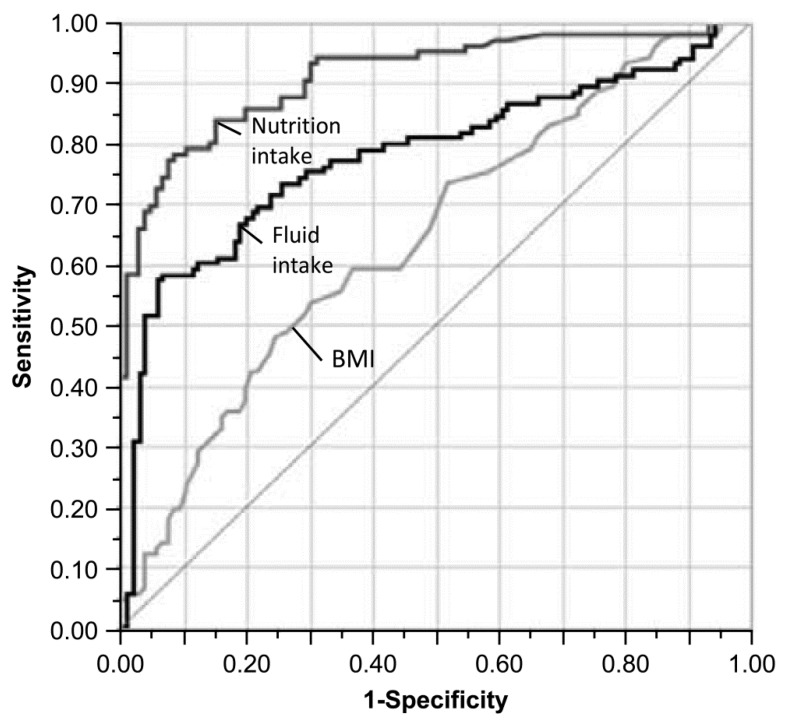
ROC curves for the amount of change during the year prior to death and the period from 24 months to 12 months prior to death.

**Table 1 ijerph-17-01823-t001:** Patients characteristics.

Item		(N = 106)	
Sex	Male	17	16%
	Female	89	84%
Mean age ± SD			
		90.1 ± 7.1	
	Male	88.9 ± 7.2	
	Female	90.4 ± 7.2	
No. of subjects according to age	≤69	0	0%
	70–74	4	4%
	75–79	4	4%
	80–84	14	13%
	85–89	21	20%
	90–94	31	29%
	95–99	24	23%
	≥100	8	8%

**Table 2 ijerph-17-01823-t002:** A comparison of values and significant differences for each parameter from 24 months prior to death until the time immediately prior to death.

Months Prior to Death	BMI (kg/m^2^)			Energy Intake (kcal)		
	Mean	Standard Deviation (SD)	*p*-Value (Comparative Value 18.5)	Mean	SD	*p*-Value (Comparative Value 830)
Immediately prior to death	16.2	3.5	<0.001 *	500	372	<0.001 *
1	16.4	3.6	<0.001 *	763	361	0.0577
2	16.6	3.5	<0.001 *	875	320	0.1471
3	16.8	3.5	<0.001 *	923	316	0.0029
4	17.0	3.3	<0.001 *	972	299	<0.001
5	17.1	3.4	<0.001 *	1002	290	<0.001
6	17.3	3.4	<0.001 *	1038	279	<0.001
7	17.5	3.4	0.0024 *	1056	294	<0.001
8	17.7	3.4	0.0188 *	1077	271	<0.001
9	17.8	3.4	0.0420 *	1091	264	<0.001
10	18.0	3.4	0.1175	1096	257	<0.001
11	18.0	3.5	0.1878	1122	249	<0.001
12	18.1	3.5	0.2175	1138	241	<0.001
13	18.2	3.5	0.3971	1125	259	<0.001
14	18.2	3.5	0.4632	1139	247	<0.001
15	18.3	3.5	0.5060	1143	252	<0.001
16	18.4	3.5	0.6659	1130	257	<0.001
17	18.5	3.4	0.9945	1142	254	<0.001
18	18.6	3.5	0.7640	1159	246	<0.001
19	18.6	3.4	0.7465	1184	235	<0.001
20	18.7	3.3	0.5554	1172	248	<0.001
21	18.9	3.4	0.2739	1151	295	<0.001
22	18.9	3.5	0.2904	1169	267	<0.001
23	18.9	3.3	0.2386	1195	233	<0.001
24	18.9	3.3	0.1721	1187	232	<0.001

One-sample *t*-test; * at *p* < 0.05, mean value < comparative value.

**Table 3 ijerph-17-01823-t003:** Amount of change in each parameter during the period immediately prior to death and during the preceding period.

Variable	△_1_		△_2_		
	Mean	SD	Mean	SD	*p*-Value
BMI (kg/m^2^)	1.9	2.0	0.9	1.8	<0.001
Energy intake (kcal)	638	382	49	204	<0.001
Fluid intake(mL)	199	243	−12	191	<0.001

**Table 4 ijerph-17-01823-t004:** AUC, cutoff value, and odds ratio of each parameter.

Variable	AUC	95% Confidence Interval		*p*-Value	Sensitivity	Specificity	Cutoff Value	Odds Ratio with 1SD
		Lower Limit	Upper Limit					
BMI (kg/m^2^)	0.645	0.574	0.721	<0.001	49%	75%	1.7	1.80
Nutritional intake (kcal)	0.912	0.875	0.953	<0.001	77%	92%	300	17.32
Fluid intake (mL)	0.775	0.720	0.848	<0.001	58%	94%	126	3.47

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
