# Peer review of "Changes in Body Mass Index, Energy Intake, and Fluid Intake over 60 Months Premortem as Prognostic Factors in Frail Elderly: A Post-Death Longitudinal Study"

_ijerph, 2020, doi:10.3390/ijerph17061823_

Round 1
Reviewer 1 Report
There are some crucial problems with the paper:
Description of patients characteristics:
How frailty is defined is lacking. What frailty instrument was used? If not used, how do we know the sample did not contain non-frail or pre-frail patients? This is a major problem and if not specified, the paper cannot be accepted as it is.
Other minor issues:
Abstract:
---- maximum of 60 months prior to death to death (typo, repeated)
Author Response
Response to Reviewer 1 Comments
We wish to express our appreciation to the Reviewer for his or her insightful comments, which have helped us significantly improve the paper.
Point 1: Description of patients characteristics:
How frailty is defined is lacking. What frailty instrument was used? If not used, how do we know the sample did not contain non-frail or pre-frail patients? This is a major problem and if not specified, the paper cannot be accepted as it is.
Response 1:
We have added the following information to Line72–Line79:
Residents of a facility covered by the Public Nursing Care Insurance, such as J Nursing Home, are required to be officially recognized as being at care-need level 3 or above as per the Long-Term Care Insurance system. A typical example of an elderly individual at care-need level 3 is someone who cannot stand or walk under one’s own power and requires full assistance for performing daily activities, such as using the toilet, bathing, and dressing, with some problematic behaviors and poor understanding [15]. All subjects in the present study met the CHS frailty criteria of Fried et al. [16] and were considered to be frail elderly individuals.
As you pointed out, it is unclear whether the subjects met the clinical diagnostic criteria; thus, we have added the following information as limitations (L260–L262):
Although we defined frail based on the Long-Term Care Insurance system in Japan, it is unclear whether all residential individuals met the clinical criteria of frail. Therefore, caution is needed while interpreting the results of our study.
Point 2: Other minor issues:
Abstract:
---- maximum of 60 months prior to death to death (typo, repeated)
Response 2:
As per your comment, we have revised it to “maximum of 60 months premortem to death” in Line18.
Reviewer 2 Report
I have read the paper entitled „Changes in body mass index, energy intake, and fluid intake over 60 months prior to death as prognostic factors in frail elderly: A post-death longitudinal study” with great interest. I found the article very interesting. I suggest the following modifications.
Comments:
17th line. Please delete “as planned”. 125th line. Please amend to “a p value <0.05 was considered significant.” Figures 1 and 2. Instead or in addition to colours it may be worthy to use dashed lines for those who are colour blind. 3.3. I struggle to understand “Amount of change from 24 months prior to death to 24 months prior to death “. Please, clarify the paragraphs and Table as currently they do not make much sense.Author Response
Response to Reviewer 2 Comments
We wish to express our appreciation to the Reviewer for his or her insightful comments, which have helped us significantly improve the paper.
Point 1: 17th line. Please delete “as planned”.
Response 1:
As per your suggestion, “as planned” in Line 17 has been deleted.
Point 2: 136th line. Please amend to “a p value <0.05 was considered significant.”
Response 2:
As per your suggestion, we have revised it to “A p-value of <0.05 was considered significant.”
Point 3: Figures 1 and 2. Instead or in addition to colours it may be worthy to use dashed lines for those who are colour blind.
Response 3:
As per your suggestion, we have modified Figures 1 and 2 as black-and-white figures.
Point 4: 3.3. I struggle to understand “Amount of change from 24 months prior to death to 24 months prior to death “. Please, clarify the paragraphs and Table as currently they do not make much sense.
Response 4:
The sentence and the notations in the table were incorrect.
We have revised the sentence to “The mean change in BMI from 12 months premortem to immediately prior to death (Δ1) and from 24 months to 12 months premortem (Δ2) were 1.9 ± 2.0 and 0.9 ± 1.8 kg/m2, respectively.” Furthermore, the notations in the table have been corrected in the same way.
Reviewer 3 Report
Thank you for the opportunity to review this interesting manuscript assessing the temporal changes in BMI, energy and fluid intake of elderly individuals living at an end of life care facility. The paper is well-presented and the findings very relevant. I suggest a thorough revision as some sentences and thoughts seem incomplete and/or unnecessarily too complex/long. The discussion section needs the most amount of work. My comments are attached to the pdf file attached. Best of luck.

Author Response
Response to Reviewer 3 Comments
We wish to express our appreciation to the Reviewer for his or her insightful comments, which have helped us significantly improve the paper.
Point 1: 55th line. This sentence seems incomplete – why is the ability to predict the time until death?
Response 1: The sentence “Regarding end-of-life care, it is important to be able to predict the time until death.” has been deleted.
Point 2: 57th line. This sentence seems a bit too long and confusing.
Response 2: We have deleted “The aim of the present study was to determine changes in BMI, energy intake, and fluid intake as well as the relationship among these parameters using data that can be easily obtained beginning 60 months prior to death to death. We collected these data using meals given to elderly individuals residing in Japanese nursing homes who died without receiving artificial nutrition and hydration (ANH).”, and have revise it to “The aim of the present study was to clarify the changes in BMI, energy intake, and fluid intake among elderly individuals residing in Japanese nursing homes who died without receiving artificial nutrition and hydration (ANH).”
Point 3: 176th line. in terms of for
Response 3: We have revised “in terms of” to “for.”
Point 4: Table3 “Amount of changes from 24 months prior to death to immediately prior to death” could use delta to make this shorter.
Response 4: The sentence and notations in the table were incorrect.
We have revised it to “The mean change in BMI from 12 months premortem to immediately prior to death (△1) and from 24 months to 12 months premortem (△2) were 1.9 ± 2.0 and 0.9 ± 1.8 kg/m2, respectively.” Furthermore, we have revised the items in the table.
Point 5: Premortem can used instead of prior to death.
Response 5: We have revised “prior to death” to “premortem”.
Point 6: 194th line. This has already been stated in results and shouldn't be there.
Response 6: We have deleted this sentence.
Point 7: 198th line. BMI is a reflection of weight changes, one can not imply that BMI 'led' to low energy intake unless you have informatiin on energy malabsorption etc . Another possibility is that patients Did you assess simptons such as tremors which may increase energy expenditure? or perhaps there were issues digesting and metabolasing energy. [sic]
Response 7: We have deleted “BMI 'led' to low energy intake,” and have revised it to “Generally, a decrease in BMI is attributed to decreased energy intake or increased relative energy expenditure due to chronic cough and increased activity. However, among the subjects in this study, a decrease in BMI preceded a decrease in food intake, suggesting that the observed BMI decrease is attributable to factors other than decreased energy intake.”(L233-236)
Point 8: 214th line. Unsure as to how this connects to the previous sentence. Is this is line with what is advised in terms of body weight?
Response 8: We have revised the Discussion section to address this point and Point9.
Point 9: 218th line. Again, perhaps you should discuss issues with malabsorption [sic]
Response 9: We have amended the Discussion section to address this point and Point8. “BMI can be easily measured and longitudinally monitored; thus, it can serve as an effective prognostic factor. Physical constitution differs depending on ethnicity, and a valid threshold for each ethnic group must be determined. It appears that in Japanese, when BMI is <18.5 kg/m2 and decreases by 1.7–1.9 kg/m2 over 1 year (approximately 10%), it can be used as a valid marker for determining a high risk of death.
The relationship of the changes in BMI with changes over time in energy intake and fluid intake levels observed in the present study differed significantly during the year prior to death and during the preceding year. The predictive ability and odds ratio increased for the measured parameters in the order of energy intake, fluid intake, and BMI.
Furthermore, While the subjects exhibited a continuously significant and irreversible decrease in the mean BMI during 9 months premortem, the mean energy intake significantly decreased immediately prior to death, Generally, a decrease in BMI is attributed to decreased energy intake or increased relative energy expenditure due to chronic cough and increased activity. However, among the subjects in this study, a decrease in BMI preceded a decrease in food intake, suggesting that the observed BMI decrease is attributable to factors other than decreased energy intake.”
Point 10: 263th line. what about strengths? you followed these poeple up for 60 months and have a very clear picture of their progression. [sic]
Response 10: We have added the following sentence: “In terms of strengths, temporal changes in each parameter were measured for up to 60 months in the present study, resulting in a more reliable prediction.”
Round 2
Reviewer 1 Report
The paper is improved and a number of concerns addressed.
However, under Methods:
I would caution the claim that "all would satisfy Fried's criteria of frailty" - when this was not carried out (the fifth criteria of Fried's tool requires testing grip strength") and to claim this retrospectively is not scientific. A recent paper below showed prevalence of frailty in Nursing Home varies widely according to tools used. I would simply acknowledge that frailty was not formally tested although authors believe according to Japanese Insurance policy, there is a high likelihood that the prevalence is high. Authors have kind of mention there is uncertainty about frailty diagnosis in discussion, so this is probably the most appropriate way to acknowledge the limitation. Otherwise, a prudent reader can ask "what tool have you used?"
J Frailty Aging. 2017;6(3):122-128. doi: 10.14283/jfa.2017.20.Prevalence of Frailty in Nursing Home Residents According to Various Diagnostic Tools.
Buckinx F1, Reginster JY, Gillain S, Petermans J, Brunois T, Bruyère O. Also what does J stand for in methods "J of Nursing Home" was mentioned. After addressing above, the paper can be accepted, but I noted some minor typo. I do not need to read it again.Author Response
We wish to express our appreciation to the Reviewer for his or her insightful comments once again, which have helped us significantly improve the paper.
Point 1: Under Methods:
I would caution the claim that "all would satisfy Fried's criteria of frailty" - when this was not carried out (the fifth criteria of Fried's tool requires testing grip strength") and to claim this retrospectively is not scientific. A recent paper below showed prevalence of frailty in Nursing Home varies widely according to tools used. I would simply acknowledge that frailty was not formally tested although authors believe according to Japanese Insurance policy, there is a high likelihood that the prevalence is high. Authors have kind of mention there is uncertainty about frailty diagnosis in discussion, so this is probably the most appropriate way to acknowledge the limitation. Otherwise, a prudent reader can ask "what tool have you used?"
Response 1:
As per your comment, “met the CHS frailty criteria of Fried et al. and” in Line 74 has been deleted.
Point 2: What does J stand for in methods "J of Nursing Home" was mentioned. [sic]
Response 2:
We have revised “J” to “a.” or “this” in Line 65 and 69.
Please see the attachment for the revised manuscript.
